# Development and Evaluation of an Internet-Based Diabetes Nutrition Education Resource

**DOI:** 10.3390/nu11061217

**Published:** 2019-05-28

**Authors:** Zhuoshi Zhang, John Monro, Bernard J. Venn

**Affiliations:** 1Department of Human Nutrition, University of Otago, P.O. Box 56, Dunedin 9054, New Zealand; zhuoshi.zhang@gmail.com; 2New Zealand Institute for Plant & Food Research Ltd., 11600 Private Bag, Palmerston North 4474, New Zealand; John.Monro@plantandfood.co.nz

**Keywords:** education, Internet, knowledge, nutrition, pre-diabetes, type 2 diabetes

## Abstract

Nutritional education for pre- and type 2 diabetes empowers individuals to make positive dietary and lifestyle choices. As the world migrates to digital devices, opportunities arise for education resources to reach a broad spectrum of society. This study aimed to develop and test the effectiveness of an electronic nutritional education resource for people with pre- and type 2 diabetes within the multi-ethnic New Zealand population. A needs assessment was conducted via ethnic-specific discussion groups (*n* = 29), followed by a population-based online survey (*n* = 448). An educational resource, including an educational video and pre- and post-questionnaires, was developed and tested online among 156 participants (17 with pre- and type 2 diabetes, 118 interested lay public and 21 health professionals). There was a strong desire to learn nutrition through simple, visual, practical, and culturally appropriate online educational resources. After interacting with the educational resource, the accuracy of identifying foods that increase blood glucose concentration improved by 17.4% (*p* = 0.013) in people with pre- and type 2 diabetes, 12.8% (*p* = 0.003) in health professionals, and 16.3% (*p* < 0.001) in interested lay public. There was an improvement among ethnic minority participants of 14.1% (*p* = 0.003). Most participants expressed intentions to make positive dietary and lifestyle choices. The electronic nutrition education resource was found to be an effective means for delivering education. It has potential to bridge the gap between the limited supply of healthcare resources and the increasing demand for diabetes nutrition education.

## 1. Introduction

Diabetes affects one in eleven adults, costing 12% of global health expenditure and will continue to rise [1,2]. While diabetes self-management education (DSME) is shown to improve patients’ diabetes knowledge [3,4,5,6,7], self-efficacy [3,4,5], dietary habits [3,6,8], and glycaemic control [3,4,5,7,8,9], only a proportion of people with diabetes receive such education due to lack of health care resources, transportation, work commitments, language barriers and financial cost [10,11].

As the world moves to electronic devices and the Internet, increasing numbers of people search for health information online [12]. In 2013, 72% of American Internet users sought health information online with 59% seeking information specifically on a medical condition, symptoms or treatment [12]. A similar uptake was found in the same year in New Zealand with 82% of New Zealand Internet users searching for online health information [13]. Meanwhile, few differences associated with this behaviour were found when comparing Europeans with non-Europeans including Hispanic, African Americans, American Indian, Asian, Hawaiian, and Mexican Americans [14,15].

Results of systematic reviews have shown that Internet based diabetes education improves patients’ knowledge and diabetes control to various degrees [16,17]. However, due to minimal patient-clinician interaction, keeping individuals engaged is crucial for the success of such education [18,19]. Engagement can be improved through providing customer-centred information and using visual aids and translated languages [15,20]. However, education content has largely been derived from text books, existing group programmes or clinicians’ experiences rather than being based on what patients wanted to know [21,22,23]. Moreover, nearly all studies were designed for people who have diabetes, with few targeted at diabetes prevention [15,24].

In order to prevent and manage type 2 diabetes, such education should be targeted at a population level, to people with or without the condition [1,25] and even to health professionals for whom nutrition and diabetes is not a speciality [26,27]. Online information can be accessed at any time, anywhere by an unlimited number of users, providing a potential solution for educating the growing number of people being diagnosed with pre- and type 2 diabetes as well as increasing the awareness of the general public, without over-stretching the healthcare system. Therefore, the aim of this research was to develop and test the effectiveness of a population-based online diabetes nutrition education resource for people with pre- and type 2 diabetes and for those who are interested in knowing its nutritional management.

## 2. Materials and Methods 

### 2.1. Study Design 

A controlled pre- and post-test design was used in this study. It was hypothesised that after participants used the Internet-based diabetes nutrition educational tool (IDNET), they would improve their knowledge of foods and beverages that increase blood glucose, and demonstrate willingness to make dietary and lifestyle changes to prevent or manage pre- and type 2 diabetes.

### 2.2. The IDNET

The IDNET included a pre-study quiz, a 20-minute educational video and a post-study quiz. The quiz consisted of three blocks of questions, with each block containing six foods and beverages randomly selected from a pool of 30–39 items within the following categories: 1. Vegetables & fruit, 2. Protein, fats, rice and wheat products, and 3. Processed foods. Participants were required to identify all foods and beverages that increase blood glucose levels (values: 1 = correct, 0 = incorrect) both before and after watching the video. A knowledge score was calculated from the correct answers. To minimise the learning effect from participating in the quiz, answers to the quiz were not provided.

### 2.3. Validation Study (Control Group)

A validation test re-test study was conducted prior to the main research. It aimed to confirm the equivalence in responses to randomly selected questions to ensure there was not bias in the questionnaire. A total of 93 participants were recruited with ethics approved by the University of Otago Ethics Committee (Reference no. D16/374).

### 2.4. IDNET Video Development (Needs Assessment)

Ethnic-specific discussion groups among people with pre- and type 2 diabetes were conducted with Māori, European, Pacific Island, Indian, and East Asian people residing in Auckland and Palmerston North, New Zealand [28]. A diabetes dietitian discussion group was also conducted in Auckland, New Zealand. The aim of these discussions was to identify ethnic-specific needs and expectations of an online nutrition diabetes educational resource. Each discussion group lasted for 60 min and was approved by University of Otago Human Ethics Committee, Reference no. 14/179 and by the Ngāi Tahu Research Consultation Committee. In addition to the discussion groups, we broadened the scope of the information-gathering exercise to include a wider demographic. This was done based on discussion-group feedback that family or community members of people with pre- or type 2 diabetes were also seeking information about the condition and its management. Hence, we instigated an online survey aimed at the general public seeking their thoughts on the information that would be useful to them in helping their family or community member with diabetes. This survey was approved by the University of Otago Human Ethics Committee, Reference no. D15/077. 

Educational topics to be included in IDNET were identified based on the results of the discussion groups and the survey. All topics were presented as a combination of videos, animations, audio, pictures and words. A variety of culture-specific foods were included to reflect the diets of ethnic groups in New Zealand. The video was pre-tested for sense and ease of use with diabetes dietitians, diabetes nurse specialists, diabetes physicians and Diabetes New Zealand. The content was in line with the Standards of Medical Care in Diabetes [29] and the New Zealand Ministry of Health guidelines [30]. The educational video included the following topics: 1)Pathology of pre-diabetes and type 2 diabetes;2)Foods and beverages that impact blood glucose;3)Meal portions and a healthy plate model;4)Meal and snack ideas;5)Exercise and nutrition tips for managing pre- and type 2 diabetes.

### 2.5. Participants and Data Collection 

This IDNET study was promoted via New Zealand primary health care organisations, Diabetes New Zealand and social media. The inclusion criteria were: All adults (age ≥ 18 years) with an interest in pre- and type 2 diabetes, who were able to read and write English, and who had access to the Internet. Participants’ demographic information was collected at the beginning of the study. Participant’s diabetes and nutrition knowledge, baseline eating habits and perceived dietary and lifestyle changes after interacting with IDNET were assessed. 

This study was approved by the University of Otago Department of Human Nutrition Academic Ethics Committee (Reference no. D16/374). A 40 second introductory video was played to all participants explaining the purpose of the study, an overview of the content and video duration. Consent to participate in the study was obtained by clicking an entry icon. The estimated length of time spent on each activity during the IDNET study is listed below in Figure 1. Participants were considered to have not completed the study if they had spent less than 15 min on the study website, because they would not have viewed a sufficient amount of the nutritional component of the video to enable learning to take effect.

### 2.6. Statistical Analysis

All data were processed using R for Macintosh (© R Foundation for Statistical Computing, 2016) and analysed using Microsoft Excel for Macintosh (Microsoft® Excel® version 15.31. Microsoft Corporation 2017, Microsoft Corporation, Redmond, Washington, US). A *p*-value of less than 0.05 was set as the cut-off for statistical significance in all analyses. 

Participants’ meal proportions were compared against the recommended healthy meal composition (i.e., ½ meal of vegetables, ¼ meal of carbohydrate, and ¼ meal of protein). A scoring system was devised based on the common denominator of 24 to quantify deviation from the recommended proportions. The absolute difference between the proportions selected by the participant and the recommended meal composition was calculated as below.
(1)$$\mathrm{Meal}\;\mathrm{deviation}\;\mathrm{score}=\left|12-\mathrm{proportion}\;\mathrm{of}\;\mathrm{vegetables}\;\times \;24\right|+\;\left|6-\mathrm{proportion}\;\mathrm{of}\;\mathrm{carbohydrate}\;\times \;24\right|+\left|6-\mathrm{proportion}\;\mathrm{of}\;\mathrm{protein}\;\times \;24\right|,$$

For example, for a participant who consumed 1/3 of a meal as vegetables, 1/3 of a meal as carbohydrate and 1/3 of a meal as protein, the meal composition score is eight.
(2)$$\Delta \mathrm{Meal}\;\mathrm{composition}=\left|12-\frac{1}{3}\;\times \;24\right|+\left|6-\frac{1}{3}\;\times \;24\right|+\left|6-\frac{1}{3}\;\times \;24\right|=\left|12-8\right|+\;\left|6-8\right|+\left|6-8\right|=4+2+2=8,$$

## 3. Results

### 3.1. Needs Assessment Result

The result of the focus group study has been published in previous Nutrients journal [28]. Six Europeans, five New Zealand Māori, four Pacific Islanders, eight East Asians and six Indians attended separate ethnic-specific discussion groups [28]. All participants had a strong desire for simple diabetes dietary education regarding culturally appropriate food, meal plans, and portion sizes [28]. Pictorial or video formats illustrated with ‘hands’ and ‘plates’ were favoured over text-based advice, particularly if that text used scientific jargon [28]. 

Auckland diabetes dietitians (*n* = 10) working in both primary and secondary care organisations took part as a discussion group. All dietitians believed there was a strong need for an online educational resource. They expressed preferences for educational information to be simple, visual, focused on carbohydrate education; and in-line with the Ministry of Health New Zealand, Diabetes New Zealand, and the Heart Foundation existing guidelines. They emphasised the importance of accurate information especially on the pathology of pre-diabetes and type 2 diabetes. The following topics were identified by diabetes dietitians and incorporated into the video: The diagnosis and physiology of pre- and type 2 diabetes; identification of carbohydrate-containing foods; importance of weight management; healthy food portions; physical activity for people with pre- and type 2 diabetes; alcohol advice; and the benefits of smoking cessation.

A total of 448 participants answered the online information-gathering survey (64 with pre- or type 2 diabetes, 312 interested lay public, and 72 health professionals). Over a third of people with pre-diabetes and type 2 diabetes reported searching for diabetes nutrition information online. This was the third most popular way after asking general practitioners and registered nurses. Accessing trusted diabetes nutritional information online was rated as the most favourable way to answer nutritional queries by all groups. Although 74% of health professional respondents believed attending a diabetes self-management education (DSME) course was beneficial, only 41% of respondents with pre- and type 2 diabetes preferred this option (*p* < 0.01). Meanwhile, 59% of respondents with pre- and type 2 diabetes preferred receiving nutrition education from their general practitioners or registered nurses, but only 37% of the health professionals who answered the survey preferred patients to do so (*p* < 0.05). Asking families, friends and pharmacists were the least preferred options (less than 20%) by participants from all groups.

Most respondents indicated that they wanted to use an online education tool as a one-off education exercise, or to use it on a weekly or monthly basis. They perceived an online education program as providing flexible learning times, written information, nutritional tips, useful pictures, videos and practical approaches. The majority of participants mentioned they would be more likely to use an online education programme if it had received positive feedback from other people with pre- or type 2 diabetes, or if it had been recommended by a health professional.

### 3.2. Validation Study

Results of the validation study (*n* = 93) showed completion of two nutrition quizzes without watching the video did not result in any knowledge gain. Additionally, randomisation of foods and beverages did not change the number of food items (out of 18 in total) correctly identified (Table 1).

### 3.3. IDNET Participants’ Characteristics

A total of 174 people out of 291 who visited the study website completed both pre- and post-video questionnaires and quizzes. Of those who completed, 156 (90%) met the inclusion criterion of having spent 15 min or more on the website (Figure 2). 

The characteristics of the survey participants are presented in Table 2. 

### 3.4. IDNET Quiz

After watching the educational video, the number of food items correctly identified as raising blood glucose increased by 17.4% (*p* = 0.013), 12.8% (*p* = 0.003) and 16.3% (*p* < 0.001) in people who had pre- and type 2 diabetes, health professionals and interested lay public respectively (Table 3). Subgroup analysis showed significant improvement in the number of food items correctly identified by 16.7% (*p* < 0.001) of the European participants and by 14.1% (*p* = 0.003) of participants with other ethnicities. 

Participants’ perceived ability to identify foods and beverages that affect blood glucose also significantly improved, with 82% of pre- and type 2 diabetes, 86% of health professionals and 49% of interested respondents reporting that they definitely knew which foods and beverages increased blood glucose as opposed to 35%, 47% and 8% respectively prior to watching the video (Figure 3).

### 3.5. Perceived Dietary and Behaviour Change

At baseline, over 85% of the participants believed people with pre- and type 2 diabetes need or may need to snack. Only 38% of health professionals, 12% of those with pre- and type 2 diabetes and 9% of interested lay public thought snacks were unnecessary. Health professionals also tended to have more regularly spaced meals than other groups, with 86% rarely skipping a meal as opposed to 64% with pre- and type 2 diabetes and 59% of interested people. 

After watching the video, the absolute difference from recommended proportions (half vegetables, quarter carbohydrate and quarter protein food) was significantly reduced in all groups (Table 4), showing intention to better align carbohydrate, protein and vegetable intake with the recommended healthy plate model than pre-video (Figure 4). 

On completion of the IDNET, 92% of the participants planned to make some dietary and/or lifestyle change such as ‘increase vegetable intake’ (52%), ‘follow the healthy plate model’ (45%), ‘have less sugar’ (44%), ‘increase physical activity’ (43%) and ‘evenly spread carbohydrate intake throughout the day’ (37%), ‘eat fewer snacks’ (33%), ‘lose weight’ (30%) and ‘increase fruit intake’ (20%). Of the 57 participants who reported not always eating regularly, 54 wanted to eat regular meals after watching the video. 

### 3.6. IDNET Feedback

Participants felt the video was easy (83%) or moderately easy (16%) to understand. They described the resource as being informative and interactive, with the quizzes enhancing the learning experience. Some mentioned that it should be given to everyone newly diagnosed with pre- and type 2 diabetes. Further suggestions included: Reduce the length of the video, have different language speakers and be able to go back to the video to check quiz answers. 

## 4. Discussion

The IDNET tool was viewed by 17 people with pre- or type 2 diabetes, 118 interested lay public, and 21 health professionals. After viewing IDNET, the ability and confidence of all groups to correctly identify foods and beverages that raise blood glucose significantly increased. This knowledge gain is consistent with previous controlled [31,32] and uncontrolled interventional studies [15,20]. As opposed to only targeting people with diabetes, the current study also included general public. To halt the rising prevalence of pre- and type 2 diabetes, population-based education on healthy eating and lifestyle is essential [1,29]. In addition, involving caregivers and family members in education promotes patients’ behaviour change [25,29,33], especially among ethnic minorities living in multi-generation and extended family households [33]. 

Similar to the results from the discussion groups [28], a general lack of understanding in nutrition for diabetes was found. Despite participants having a relatively high level of education, it is concerning that at baseline only 35% of participants with pre- and type 2 diabetes, 48% of health professionals and 7% of those interested thought that they definitely knew which foods and beverages increase blood glucose. These identified knowledge deficits and emphasised the need for a population-based nutrition education resource. A significant knowledge improvement was shown in all groups after viewing the IDNET, even among health professionals who have been found to pass the information to their clients [26,27]. People with pre- and type 2 diabetes perceived that they had a better knowledge base than those without the condition although in reality, they did not outperform people without diabetes in the nutrition quiz. Indeed, although people with the condition may have received nutrition information from various sources, some information may have been inaccurate or even conflicting [28,34,35]. This in turn led to misconceptions, confusion and mistrust of advice [28,34,36]. 

Motivation and engagement is crucial when the education is delivered by distance via electronic devices. A strength of this study was its strong consumer component. In many studies, resources have been designed based on text books, existing group programmes or the experience of clinicians [21,22,23]. The co-design of the IDNET incorporated ideas expressed by people with pre- and type 2 diabetes of different ethnicities, health professionals and lay public interested in this condition. Ease of use and the provision of information specific to peoples’ needs increases consumer satisfaction and promotes learning outcomes [18,19]. Pre-learning tests further enhance learning engagement and knowledge retention [37]. Participants found that the pre-video quizzes addressed their knowledge gap and aroused their interest, which enhanced the study result. Future researches should investigate whether this knowledge gain and motivation to change will ultimately transfer to sustained behaviour change. 

The use of video, animation, pictures, text and voice catered for people with different learning styles [38]. It promoted learning outcomes for people with lower health literacy and for those with limited language proficiency [39]. Non-European participants specifically requested such features during group discussions [28]. Evidence supporting the success is the significant knowledge gain in non-European participants. 

A limitation of the study was a relatively high dropout rate before watching the video. Although the resource was promulgated through social media and health care organisations, the ability to reach a large proportion was restricted by the study timeframe. Additionally, having to complete a participant characteristic survey and a nutrition quiz before the video may have weakened participants’ motivation. Although the video was only twenty minutes in duration, that may have been too long for some people. A possible solution to this would be to split the video into 5–10 min sessions to lower the initial time requirement. Although both European and non-European participants showed significant knowledge improvement after using the IDNET, the resource was only presented in English, limiting comprehension for non-English speakers. However, translating into other languages is possible if it is thought that this would be useful in the future. Generalisability may also be limited as the study population was relatively young with a high education level. About 80% of the respondents were female, suggesting the need to engage and encourage men to participate in these types of activities. Furthermore, we did not measure an effect of having used the resource on diabetes management, as the purpose at this stage of the development was to assess its viability as an educational tool. Following refinement of the resource based on the results of this component of the work, and the translation into other languages, the resource should be tested for efficacy both in conjunction with formal diabetes management training and as a stand-alone resource for carers and for people with pre- and type 2 diabetes that have not attended formal training. Efficacy could include a qualitative perspective regarding the confidence with which carers and people with pre- and type 2 diabetes approach nutritional management; and with a metabolic component in which measures of glycaemic control are assessed.

## 5. Conclusions

It was found that the electronic nutrition education resource improved participants’ knowledge as well as their intentions toward healthy eating and lifestyle. Further studies can be designed to measure whether short 5–10 min videos presented in multiple languages may improve participation, and also if this learning exercise leads to actual behaviour change and improvement in patients’ diabetes control. 

## Figures and Tables

**Figure 1 nutrients-11-01217-f001:**
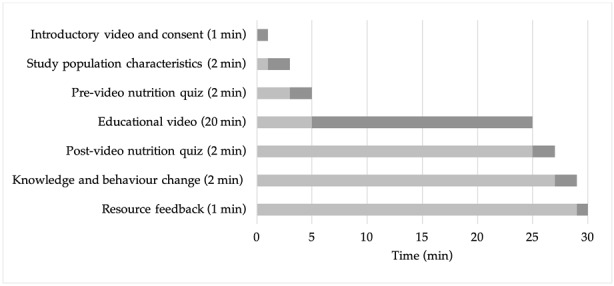
Study activities and time requirement. Dark grey bars represent time required to complete each section.

**Figure 2 nutrients-11-01217-f002:**
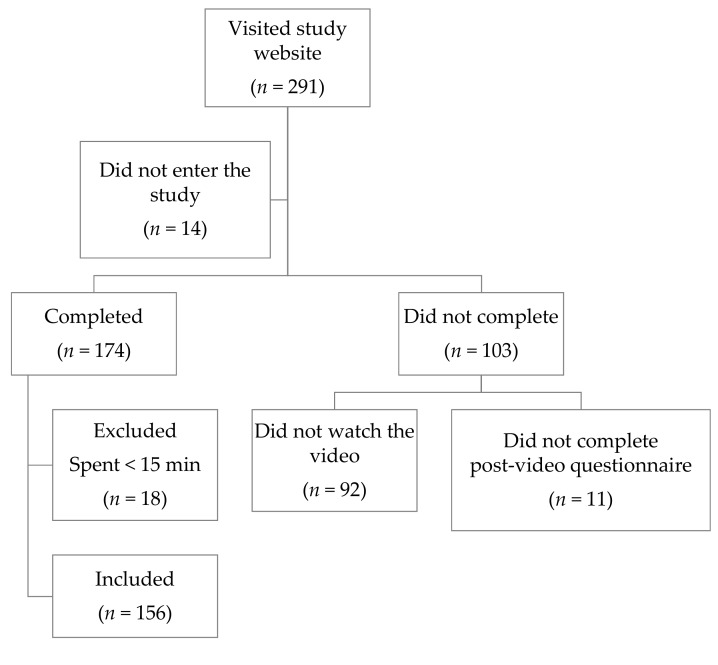
Enrolment and participation results flowchart.

**Figure 3 nutrients-11-01217-f003:**
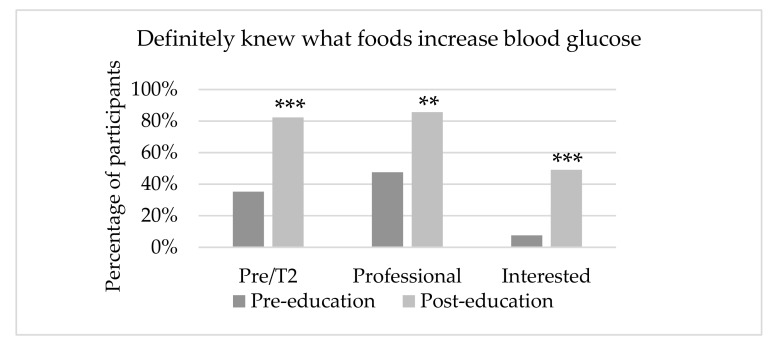
Participants who perceived they knew what foods increase blood glucose. ** *p* = 0.002, *** *p* < 0.001.

**Figure 4 nutrients-11-01217-f004:**
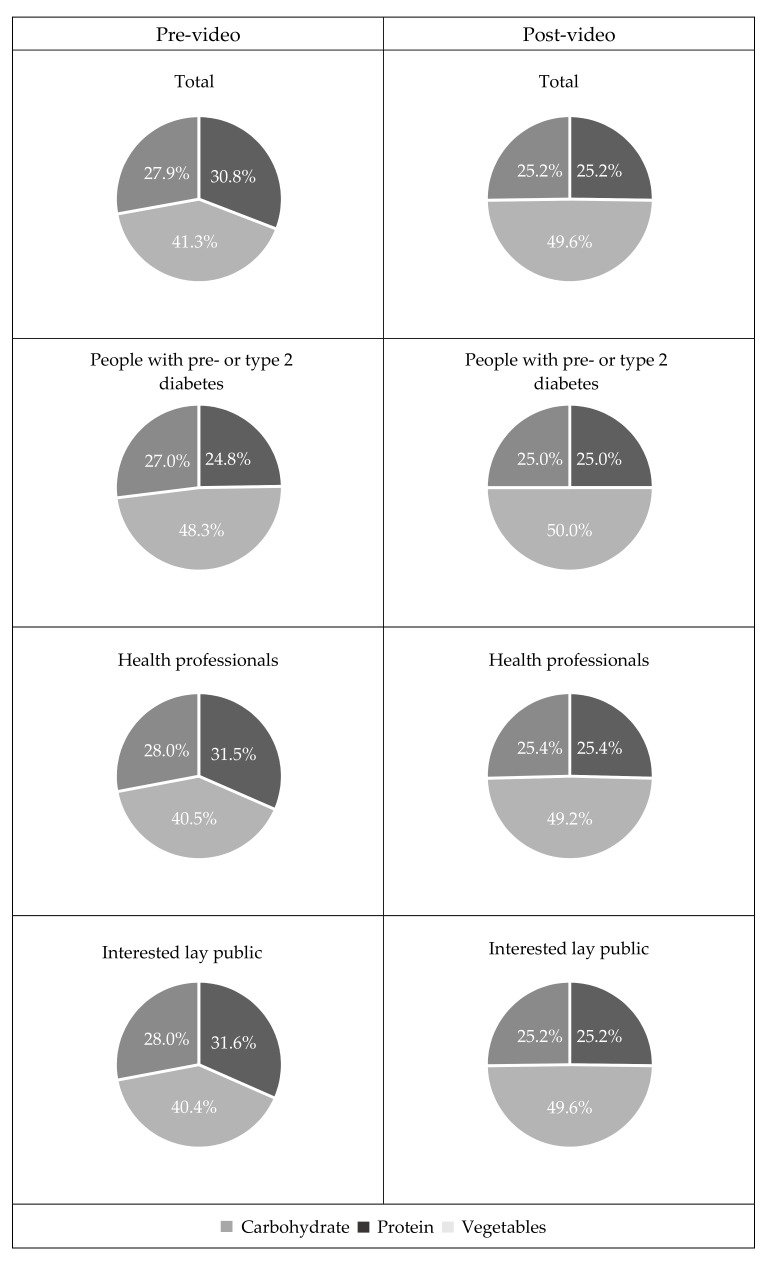
Proportion of carbohydrate, protein and vegetables at main meals.

**Table 1 nutrients-11-01217-t001:** Numbers of food items correctly identified in the validation study (*n* = 93).

	Test	Re-Test	∆Score (95% CI)	*p*-Value
Validation	12.7	12.9	5.4% (−1.7%, 12.3%)	0.394

CI, confidence interval.

**Table 2 nutrients-11-01217-t002:** Characteristics of participants (*n* = 156).

Characteristics	*n* (%)
Age, average (range) years	34.3 (19–76)
Female, n (%)	126 (81%)
Ethnicity, n (%)	
European	110 (70%)
Non-European	56 (30%)
Education, university, *n* (%)	133 (85%)
Average years of diagnosis, years	
Pre-diabetes	2.7
Type 2 diabetes	9.1
Pre- or type 2 diabetes, *n* (%)	17 (11%)
Health professionals, *n* (%)	21 (14%)
Interested lay public, *n* (%)	118 (75%)
At risk of type 2 diabetes	7 (5%)
Look after people with type 2 diabetes	6 (4%)
Have type 1 diabetes	2 (1%)
Study or research diabetes	27 (17%)
Personal interest	76 (49%)

**Table 3 nutrients-11-01217-t003:** Numbers of food items correctly identified pre- and post-video.

	n	Pre-video	Post-video	∆ Knowledge (95% CI)	*p*-Value
People with pre- or type 2 diabetes	17	13.2	15.1	17.4% (5.3%, 29.7%)	0.013
Health professionals	21	14.8	16.4	12.8% (5.1%, 20.6%)	0.003
Interested lay public	118	13.6	15.2	16.3% (11.1%, 21.5%)	<0.001
European	110	13.8	15.5	16.7% (11.7, 21.7%)	<0.001
Non-European	46	13.5	14.9	14.1% (5.9%, 22.3%)	0.003
Total	156	13.7	15.4	16.0% (4.3, 20.2%)	<0.001

CI, confidence interval.

**Table 4 nutrients-11-01217-t004:** Comparison of meal composition to healthy plate model.

	n	Δ Meal Composition Pre-video (95% CI)	Δ Meal Composition Post-video (95% CI)	*p*-Value
People with pre- or type 2 diabetes	17	3.6 (0.9–6.4)	0.0 (0.0, 0.0)	0.018
Health professionals	21	5.4 (3.1–7.8)	0.4 (−0.4, 1.1)	<0.001
Interested lay public	118	5.9 (4.9–6.9)	0.3 (0.1, 0.6)	<0.001
Total	156	5.6 (4.7–6.5)	0.8 (0.1, 0.5)	<0.001

CI, confidence interval.

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
