# Peer review of "Development and Evaluation of an Internet-Based Diabetes Nutrition Education Resource"

_nutrients, 2019, doi:10.3390/nu11061217_

Round 1

Reviewer 1 Report

Overall, Author tired to develop new protocol for treatment of diabetes.   

I have 1 minor point.

In Figure 3, Please put title in Y axis. And, please put statistical bar in right position.

Author Response

We would like to thank the reviewer for the time and consideration of our manuscript.

Reviewer comment: In Figure 3, Please put title in Y axis.

Author response: Thank you, we have added a title for Y axis

Reviewer comment: In Figure 3 please put statistical bar in right position.

Author response: We are unsure what the problem is here. We are indicating significant differences between the pre- and post-educational results among three participant groups; people with pre- and type 2 diabetes (left hand pair of columns); health professionals (centre pair of columns); and people interested in learning about diabetes (right hand pair of columns). The positions of the statistical bars line up as intended on our version of the Figure and we wonder whether it is a formatting issue in the reviewers version  

Reviewer 2 Report

I read the manuscript with a lot of interest. Offering reliable and valid nutrition-based information specifically for individuals with chronic condition is a significant healthcare need today. The authors describe an education resource that may be able to meet some of this need however, more detailed information should be provided in the manuscript regarding the development of IDNET. I have outlined a few of my pressing concerns below:

(1) The methods section should also discuss the development process of IDNET. How was it informed including details of the content so that readers can also assess its applicability.

(2) The discussion of results should also include information regarding next steps in terms of implementing this resource because currently there is no long-term efficacy data on this intervention. This is a pretty big limitation as a result of which this study would be better presented as a pilot attempt.

Author Response

Reviewer comment (1): The methods section should also discuss the development process of IDNET. How was it informed including details of the content so that readers can also assess its applicability.

Author response: Thank you for your recommendation. We have substantially added more detail of the way in which the development of the resource was undertaken in section 2.4 of the Methods as follows:

Ethnic-specific discussion groups among people with pre- and type 2 diabetes were conducted with Māori, European, Pacific Island, Indian, and East Asian people residing in Auckland and Palmerston North, New Zealand [28]. A diabetes dietitian discussion group was also conducted in Auckland, New Zealand. The aim of these discussions was to identify ethnic-specific needs and expectations of an online nutrition diabetes educational resource. Each discussion group lasted for 60 minutes and was approved by University of Otago Human Ethics Committee, Reference no. 14/179 and by the Ngāi Tahu Research Consultation Committee. In addition to the discussion groups, we broadened the scope of the information-gathering exercise to include a wider demographic. This was done based on discussion-group feedback that family or community members of people with pre- or type 2 diabetes were also seeking information about the condition and its management. Hence, we instigated an online survey aimed at the general public seeking their thoughts on the information that would be useful to them in helping their family or community member with diabetes. This survey was approved by the University of Otago Human Ethics Committee, Reference no. D15/077. Educational topics to be included in IDNET were identified based on the results of the discussion groups and the survey. All topics were presented as a combination of videos, animations, audio, pictures and words. A variety of culture-specific foods were included to reflect the diets of ethnic groups in New Zealand. The video was pre-tested for sense and ease of use with diabetes dietitians, diabetes nurse specialists, diabetes physicians and Diabetes New Zealand.

We have also given more detail regarding the applicability of the resource by describing the diabetes dietitians discussion feedback in the Results section as follows:

Auckland diabetes dietitians (n=10) working in both primary and secondary care organisations took part as a discussion group. All dietitians believed there was a strong need for an online educational resource. They expressed preferences for educational information to be simple, visual, focused on carbohydrate education; and in-line with the Ministry of Health New Zealand, Diabetes New Zealand, and the Heart Foundation existing guidelines. They emphasised the importance of accurate information especially on the pathology of pre-diabetes and type 2 diabetes. The following topics were identified by diabetes dietitains and incoporated into the video: the diagnosis and physiology of pre- and type 2 diabetes; identification of carbohydrate-containing foods; importance of weight management; healthy food portions; physical activity for people with pre- and type 2 diabetes; alcohol advice; and the benefits of smoking cessation.

Reviewer comment (2): The discussion of results should also include information regarding next steps in terms of implementing this resource because currently there is no long-term efficacy data on this intervention.

Author response: This is an excellent point and we agree that our study examined short-term knowledge gain without long-term efficacy data. We have expanded on this as a limitation in the last paragraph of discussion as follows:

Furthermore, we did not measure an effect on diabetes management of having used the resource as the purpose at this stage of the development was to assess its viability as an educational tool. Following refinement of the resource based on the results of this component of the work, and the translation into other languages, the resource should be tested for efficacy both in conjunction with formal diabetes management training and as a stand-alone resource for carers and for people with pre- and type 2 diabetes that have not attended formal training. Efficacy could include a qualitative perspective regarding the confidence with which carers and people with pre- and type 2 diabetes approach nutritional management; and with a metabolic component in which measures of glycaemic control are assessed.

Reviewer comment (2): This is a pretty big limitation as a result of which this study would be better presented as a pilot attempt.

The study was designed and tested as a one-off education exercise therefore we would not regard this as a pilot, rather as a completed project describing the development and testing of a resource with a gain in education as the outcome.